# Siponimod Attenuates Neuronal Cell Death Triggered by Neuroinflammation via NFκB and Mitochondrial Pathways

**DOI:** 10.3390/ijms25052454

**Published:** 2024-02-20

**Authors:** Mikel Gurrea-Rubio, Qin Wang, Elizabeth A. Mills, Qi Wu, David Pitt, Pei-Suen Tsou, David A. Fox, Yang Mao-Draayer

**Affiliations:** 1Division of Rheumatology, Department of Internal Medicine, University of Michigan, Ann Arbor, MI 48109, USA; mikelg@med.umich.edu (M.G.-R.); qiw@med.umich.edu (Q.W.); ptsou@med.umich.edu (P.-S.T.); dfox@med.umich.edu (D.A.F.); 2Department of Neurology, University of Michigan Medical School, Ann Arbor, MI 48109, USA; qinwang@umich.edu (Q.W.);; 3Autoimmunity Center of Excellence, University of Michigan Medical School, Ann Arbor, MI 48109, USA; 4Department of Neurology, Yale Medicine, New Haven, CT 06473, USA; david.pitt@yale.edu; 5Multiple Sclerosis Center of Excellence, Oklahoma Medical Research Foundation, Oklahoma City, OK 73104, USA

**Keywords:** multiple sclerosis (MS), human neural stem/progenitor cells (hNSCs/NPCs), MAPK/NFκB, oxidative stress, neuroprotection

## Abstract

Multiple sclerosis (MS) is the most common autoimmune demyelinating disease of the central nervous system (CNS), consisting of heterogeneous clinical courses varying from relapsing-remitting MS (RRMS), in which disability is linked to bouts of inflammation, to progressive disease such as primary progressive MS (PPMS) and secondary progressive MS (SPMS), in which neurological disability is thought to be linked to neurodegeneration. As a result, successful therapeutics for progressive MS likely need to have both anti-inflammatory and direct neuroprotective properties. The modulation of sphingosine-1-phosphate (S1P) receptors has been implicated in neuroprotection in preclinical animal models. Siponimod/BAF312, the first oral treatment approved for SPMS, may have direct neuroprotective benefits mediated by its activity as a selective (S1P receptor 1) S1P1 and (S1P receptor 5) S1P5 modulator. We showed that S1P1 was mainly present in cortical neurons in lesioned areas of the MS brain. To gain a better understanding of the neuroprotective effects of siponimod in MS, we used both rat neurons and human-induced pluripotent stem cell (iPSC)-derived neurons treated with the neuroinflammatory cytokine tumor necrosis factor-alpha (TNF-α). Cell survival/apoptotic assays using flow cytometry and IncuCyte live cell analyses showed that siponimod decreased TNF-α induced neuronal cell apoptosis in both rat and human iPSCs. Importantly, a transcriptomic analysis revealed that mitochondrial oxidative phosphorylation, NFκB and cytokine signaling pathways contributed to siponimod’s neuroprotective effects. Our data suggest that the neuroprotection of siponimod/BAF312 likely involves the relief of oxidative stress in neuronal cells. Further studies are needed to explore the molecular mechanisms of such interactions to determine the relationship between mitochondrial dysfunction and neuroinflammation/neurodegeneration.

## 1. Introduction

Multiple sclerosis (MS) is an autoimmune disease of the central nervous system (CNS) and consists of heterogeneous clinical courses varying from relapsing-remitting MS (RRMS) to progressive disease, such as primary progressive MS (PPMS) and secondary progressive MS (SPMS). It is a leading cause of non-traumatic neurologic disability in young adults. MS is characterized by multifocal inflammation and the breakdown of the myelin sheath, which is caused by inflammatory cells in lesions in the CNS [1], and increased levels of inflammatory cytokines such as tumor necrosis factor-α (TNFα) in the cerebrospinal fluid (CSF) [2,3]. In contrast to RRMS, where several highly effective disease-modifying therapies (DMTs) have been approved to target the inflammatory aspect of the disease, there are very limited options for the neurodegeneration that is thought to be a driver of the continuous decline in progressive patients [4].

Siponimod (MAYZENT^®^, BAF312) is the first oral DMT approved for secondary progressive MS [5]. It is a sphingosine-1-phosphate receptor-1 (S1P1)- and S1P5-specific agonist which acts as a functional antagonist. In addition to its mode of action of preventing egress of proinflammatory lymphocytes out of lymph nodes, siponimod/BAF312 may have direct beneficial effects in the CNS mediated by S1P1 and/or S1P5 [6]. The S1P receptors S1P1, S1P2, S1P3, and S1P5 are expressed by neural progenitor cells (NPCs), while neurons predominantly express S1P1 and S1P3 [7,8,9]. Siponimod/BAF312 does not require a phosphorylation step in vivo. Fingolimod, the first approved oral S1P modulator, needs to be phosphorylated to the active fingolimod-phosphate (FTY-P), which works on S1P1, 2, 3, 4, and 5 to have anti-inflammatory properties. S1P receptor modulators are lipophilic and can cross the blood–brain barrier (BBB) into the CNS, and brain atrophy was reduced by the treatment in MS [10,11,12].

To gain a better understanding of the potential neuroprotective effects of siponimod/BAF312, we used both rat neurons and human-induced pluripotent stem cell (iPSC)-derived neurons to model the MS neuronal system in which neuroinflammation was induced by TNF-α, and performed transcriptomic, flow cytometry, and live-cell analysis to gain mechanistic insight. 

## 2. Results

### 2.1. S1P1 Expression in Demyelinating Lesions of MS Patient Autopsy Tissue

We used autopsy tissue from the MS brain bank to measure S1P1 receptor expression in MS brain lesions. Figure 1A shows a low magnification of a chronic active MS lesion with the green square at the lesion border and the red square at the lesion center. Streaks of myelin labeled with MBP (brown) are visible at the lesion border with high magnification, as depicted in Figure 1B. A cortical type III MS lesion is shown in Figure 1C,D, at low and high magnification, respectively. S1P1 was expressed in astrocytes at the border of active white matter lesions and to a lesser extent at the lesion center (Figure 1E,F). In cortical lesions, S1P1 is expressed predominantly by neurons within the lesion (Figure 1G,H). As seen in Figure 1H, the expression of the S1P1 receptor in cortical demyelinating lesions was abundant in neurons.

### 2.2. Siponimod/BAF312 Dose-Dependently Enhanced Neuronal Survival in Rat Primary Cultured Cortical Neurons

Rat cortical neurons were isolated and grown in a 96-well plate with neural basal medium and then transfected with EGFP and imaged for 10 days under treatment with siponimod/BAF312 at different doses ranging from 0.1 nM to 0.5 nM and 1.0 nM, as indicated in Figure 2. After following thousands of individual EGFP-labelled rat cortical neurons, the cumulative hazard curves were plotted, displaying the risk of natural death for the cortical neurons under siponimod/BAF312 treatment compared to the vehicle DMSO alone without any stressor. Our data showed that siponimod/BAF312 dose-dependently decreased the risk of cortical neuron death with HR 0.84, 0.81, and 0.78 and *p* values of 0.007, 0.002, and 2 × 10^−5^ for the cumulative risk of neuron death, respectively, up to 250 h recording time. Our results indicate that siponimod/BAF312 could enhance neuronal survival in rat primary cultured cortical neurons.

### 2.3. Siponimod/BAF312 Decreased Apoptosis in Human-Induced Pluripotent Stem Cells and Neural Stem Progenitor Cells (hiPSCs-NSCs/NCPs) by Live-Cell Analysis

We next examined the pro-survival effect of siponimod/BAF312 on human neuronal cells under inflammatory condition with TNFα. TNFα has been shown to be elevated in the CSF of MS patients [2,3], and is shown to be involved in the progression of MS [13,14]. We took advantage of the IncuCyte^®^ live imaging system to visualize cell death in hiPSCs-NSCs/NPCs, using a caspase-3/7 green assay which generates a green, fluorescent signal upon the activation of apoptotic pathways [15,16]. Human iPSCs-NSCs/NPCs cells were cultured in a 96-well plate and treated with varying concentrations (0.01, 0.1, 0.5, 1.0, 5.0 nM) of siponimod/BAF312 under TNFα (20 ng/mL) treatment, in media without growth factors. As seen in Figure 3A, TNFα increased the apoptosis of hiPSCs-NSCs/NPCs cells in a dose-dependent manner. Importantly, with the exception of the lowest dose (0.01 nM), siponimod/BAF312 significantly decreased the number of apoptotic cells per image following TNFα exposure in a dose- and time-dependent manner, as indicated by green fluorescence released by activated caspase-3/7, compared to TNFα alone (Figure 3B,C). Moreover, the effect of siponimod/BAF312 on neuronal survival was sustained until the end of the experiment (Figure 3C). We further confirmed the results using flow cytometry analysis.

### 2.4. Siponimod/BAF312 Reduced Apoptosis in Human-Induced Pluripotent Stem Cells and Neural Stem Progenitor Cells (hiPSCs-NSCs/NCPs) by Flow Cytometry Analysis

Human iPSCs-NSCs/NPCs cells were treated with siponimod/BAF312 at concentrations of 0, 10, 100, and 1000 pM with or without TNFa for 48 h (Appendix A). Live cells were double-negative for both Annexin V and 7AAD, and relative cell death or survival was calculated compared to that of relevant vehicle controls (DMSO). There was no significant effect on survival in the absence of TNFα (Appendix A), but there was a statistically significant decrease in the number of apoptotic hiPSCs-NSCs/NPCs cells with siponimod/BAF312 treatment at concentrations of 10 pM and 100 pM compared to vehicle alone (DMSO) under the TNFa treatment (20 ng/mL) (Appendix A), confirming the relevance to human neurons.

### 2.5. Transcriptomic Analysis of TNFα-Treated Human iPSCs-NSCs/NPCs

RNA-Seq analysis was performed on human neuronal iPSCs-NSCs/NPCs cells in the absence or presence of TNFα. TNFα treatment led to the differential expression of 3556 genes in hiPSCs-NSCs/NPCs cells, with 1819 genes upregulated and 1737 genes downregulated (Figure 4). These included genes involved in TNF signaling pathways, cytokine-cytokine receptor interaction, apoptosis, MAPK/NFκB signaling pathways (Appendix A), and mitochondria respiratory chain complex metabolic pathways (Figure 5).

### 2.6. Transcriptomic Analysis of Siponimod/BAF312 Treatment of TNFα-Stressed Human iPSCs-NSCs/NPCs Identifies Neuroprotective Effects via Oxidative Phosphorylation Pathways

We identified 580 genes that were significantly differentially expressed by the treatment of siponimod/BAF312 (0.1 nM) under TNFα (20 ng/mL) stress in hiPSCs-NSCs/NPCs cells, including 391 upregulated genes and 189 downregulated genes (Figure 6B). Further Kyoto Encyclopedia of Genes and Genomes (KEGG) pathway enrichment analysis revealed that the differentially expressed genes in response to siponimod/BAF312 treatment genes were enriched in targets important for oxidative phosphorylation, neuroinflammation, and neurodegenerative diseases, such as Parkinson’s disease, Alzheimer’s disease, and Huntington’s disease (Figure 6D,E and Figure 7). In addition, gene ontology analysis showed that siponimod/BAF312 treatment under TNFα neuroinflammation affected genes in a wide spectrum of cellular components, biological processes, and molecular functions known to play critical roles in neuroinflammation (Figure 6, Figure 7 and Figure 8). Not only can siponimod/BAF312 reverse some of the gene expression changes stemming from TNFα-induced MAPK/NFκB signaling changes, but also it affects mitochondrial oxidative phosphorylation signaling pathways by increasing *NDUFA2* and *NDUFB8* and decreasing *MT-ND4L*, *MT-ND6, MT-CO3*, and *MT-ATP6* (Figure 6 and Figure 7).

## 3. Discussion

Sphingosine-1-phosphate (S1P) is a bioactive sphingolipid that regulates a wide range of physiological processes, including lymphocyte recirculation, cardiac function, and the maintenance of the BBB [6,17,18]. We first showed that the S1P1 receptor was highly expressed in cortical demyelinated lesions in autopsied human MS tissue (Figure 1). O’Sullivan, C. et al. [19] showed a dual S1PR1/S1PR5 action by siponimod/BAF312 that attenuated demyelination in organotypic slice cultures. It has also been reported that rat NPCs have abundant S1P1/3/5 [20,21] and mouse NPCs express S1P1/2/3/4/5 [22]. Since S1P-receptors are ubiquitinated and subsequently degraded when exposed to FTY720-P or siponimod/BAF312, the mechanism of action of these drugs involves functional antagonism by persistent internalization and enhanced degradation of the S1P-receptor [23]. Next, we utilized the IncuCyte^®^ live imaging system to assess cell death in human hiPSCs-NSC/NPCs, which confirmed our apoptosis studies carried out using flow cytometry, showing that siponimod/BAF312 dose- and time-dependently decreased the number of apoptotic hiPSCs-NSCs/NPCs cells under the neuroinflammatory stressor TNFα (Figure 4 and Appendix A). Despite the robust anti-apoptotic effects of siponimod/BAF312, the mechanism of action by which this drug decreases apoptosis is unknown.

Our comprehensive transcriptomic RNA-seq analysis in human iPSCs-NSCs/NPCs now provides mechanistic insight into siponimod/BAF312′s role in reducing TNFα-induced neuronal death. Our data show that TNFα increased *CCL2/MCP1*, *CCL20*, *NFkB2*, *CXCL5/6*, and *TBX22* gene expression in neurons, which correlates with our previous studies showing that MCP1/CCL2 is increased in both RRMS and SPMS when compared to nonprogressive benign MS (BMS) [24]. Importantly, apoptotic and MAPK/NFκB pathways were significantly upregulated in neurons by TNFα treatment (Figure 5, Figure 6 and Appendix A). Moreover, siponimod/BAF312 showed a significant increase in mitochondrial oxidative phosphorylation pathways (Figure 6, Figure 7 and Figure 8), suggesting that the regulation of genes such as *MT-ND4L/4/5/6*, *MT-ATP6*, *NDUFA2*, and *NDUFB8* may reduce apoptotic cell death by increasing resilience to oxidative stress damage from neuroinflammation triggered by TNFα.

It has become increasingly clear that dysfunctional mitochondria are important contributors to the damage and loss of both axons and neurons [25]. This concept is also supported by findings that multiple genes, including *NDUFA2* and *TBX6*, are differentially expressed and involved in the negative regulation of cell death, nervous system development, and ATP metabolic processes in MS cortical neurons [26]. We recently showed that mitochondrial complex associated genes *NDUFS4* and *ATP5* had increased expression in RRMS and SPMS, as compared to nonprogressive BMS [27]. Previous studies and evidence support a complex link between mitochondrial dysfunction and neuroinflammation as causes of neurodegenerative diseases [28]. Importantly, activated glia release proinflammatory factors such as IL-18, IL-1β, and IL-6, which aggravate mitochondrial damage, form a vicious cycle of mitochondrial dysfunction and neuroinflammation which can initiate the NFκB-mediated nuclear signaling cascade. In addition, mitochondrial-derived Cytochrome C can activate the MAPK-JNK signaling cascade, leading to neuroinflammation. We previously also reported that dimethyl fumarate (DMF) had a neuroprotective effect by increasing Nrf2 [29], and that this result was supported by findings that multiple genes, including *MAPK3/4/6/7/9/15* and *NFKBIZ/A*, were differentially expressed. Such gene products are involved in MAPK/NFκB pathways in MS cortical neurons [26]. Our current data and other results [25,26,27,28] suggest that cross-talk between mitochondrial dysfunction and neuroinflammation may promote neurodegeneration in MS and might explain how in acute MS lesions, intra-axonal mitochondria are damaged by inflammation-derived reactive oxygen species (ROS) and nitric oxide (NO), which leads to mitochondrial dysfunction and axonal injury.

In summary, our data support a neuroprotective effect of siponimod/BAF312 in rat cortical neuron and human neuronal cells using human iPSCs-NSCs/NPCs. Our study provides evidence of the MAPK/NFκB signaling pathway, mitochondrial oxidative phosphorylation, and cytokine–cytokine receptor interaction enhancing resilience against inflammatory oxidative stress in mediating the neuroprotective effect of siponimod/BAF312 (Figure 9). Thus, interventions directed at reversing the consequences of neuroinflammation-induced neuronal death could be assessed as a therapeutic option for progressive MS.

## 4. Materials and Methods

### 4.1. S1P1 Expression in Cortical Demyelinating Lesions of an MS Brain

S1P1 receptor expression was assessed in lesions from the autopsy brain tissue of an MS patient obtained from the brain bank. In situ hybridization was performed on frozen and formalin-fixed tissue using RNAscope^®^ Technology (Advanced Cell Diagnostics), which allows for high-signal amplification with low background noise from non-specific hybridization. The probe for the CNS-relevant sphingosine1-phosphate receptor S1P1 was made of a pool of 20 complimentary sequence oligonucleotides to hybridize within a 1 kb region; the probes were labeled with horseradish peroxidase. We used the following probes: Hs-S1P1; NM_001400.4; probe region 557–1652, Hs-POLR2A (RNA polymerase 2a; probe region 2514–3433; positive control), and DapB (Bacillus subtilis; probe region 414–862; negative control). Formalin-fixed paraffin-embedded (FFPE) brain sections were de-paraffinized in xylene and gradually dehydrated in EtOH. Sections were pretreated with hydrogen peroxide to quench endogenous peroxidase activity, heated in buffer to retrieve mRNA and incubated with proteinase to digest protein for the enhanced accessibility of mRNA. Sections were hybridized with the respective probes and the mRNA-bound probes were subsequently amplified and reacted with 3,3′-diaminobenzidine (DAB) (Sigma, Burlington, MA, USA) for visualization. The stained sections were counterstained with hematoxylin, dehydrated, and mounted.

### 4.2. Preparation of Rat Neural Progenitor Cells (NPCs)

Rat NPCs were isolated as previously described [29,30,31,32]. In brief, rat embryonic E15-18 pups were decapitated, and the whole brain without cerebellum was dissected and washed with fresh HBSS solution, then digested with DNase I and filtered through a cell strainer (70 µm, Falcon). Cells were plated onto dishes pre-equilibrated with self-renewal media (SRM) containing DMEM, Neurobasal-A media, 2-mercaptoethanol, Chick Embryo Extract (CEE), 1× N-2 supplement, 2× B-27 supplement, FGF (20 ng/mL), EGF (20 ng/mL), and Penicillin-Streptomycin. Rat NPC preparation and growth were carried out as described previously [29,32].

### 4.3. Kaplan–Meier Survival Analysis for Rat Cortical Neurons

Rat cortical neurons were isolated as previously described and grown in 96-well plates with neural basal media [33,34]. The cortical neurons were transfected with EGFP for 1–2 h using Lipofectamine 2000 from Invitrogen (Thermo Fisher, Waltham, MA, USA), then treated with Siponimod/BAF312 at different concentrations. Imaging began the same day and continued for 10 days, at least once per day. Real-time survival analysis was performed using automated fluorescent microscopy to image and follow thousands of individual rat cortical neurons over a period of 10 days. GFP-labeled neurons were chosen from the first image and followed until there was an abrupt loss of fluorescence or disintegration of the cell, both of which have proven reliable as traditional markers of cell death [33]. Using Kaplan–Meier survival analysis, the cumulative hazard curves were plotted, displaying the risk of death for the cortical neurons under various conditions. Hazard Ratio (HR) and associated *p* value were calculated using COX proportional hazards analysis. HR < 1 denotes protection.

### 4.4. Differentiation of Human-Induced Pluripotent Cells (iPSCs) into Forebrain-Specific Neural Stem Cells (NSCs)/Neural Progenitor Cells (NPCs) (hiPSCs-NSCs/NPCs)

In this study, we used an induced human neuronal cell line (hiPSCs-NSC/NPCs) for various mechanistic studies. Our human iPSCs-NSCs/NPCs cell line was provided by Dr. Jack Parent’s laboratory at the University of Michigan. In brief, human-induced pluripotent cells (iPSCs) were generated from peripheral blood of a healthy white female at age 59 (purchased from Interstate Blood Bank, Inc., Memphis, TN, USA) using the Erythroid Progenitor Reprogramming Kit following a manufacturing protocol (Stemcell technologies, Cambridge, MA, USA) to isolate and expand erythroid progenitor cells and with subsequent reprogramming to induce pluripotent stem cells (iPSCs) [35]. Using these iPSCs, the neural stem cells were generated with dual SMAD inhibition following a published protocol [36,37]. First, the iPSCs were dissociated with Accutase (Thermo Fisher Scientific, Inc.) and plated on Matrigel (Thermo Fisher Scientific, Inc.)-coated 6-well plates in TeSR-E8 supplemented with 10 µm Y27632 (Dihydrochloride, Cambridge, MA, USA). The next day, the media were changed to neural induction media which contained neural maintenance media (3N) supplemented with 1 μM Dorsomorphin and 10 μM SB431542. Cells were maintained in neural induction media with daily media changes for 10–12 days to form a monolayer neuroepithelium sheet. The neuroepithelium sheet was dissociated mechanically using a comb and re-plated on Matrigel-coated 6-well plates in 3N medium supplemented with 20 ng/mL bFGF2 for an additional 4–8 days to expand NSCs/NPCs. When rosettes appeared on the neural epithelium sheet, they were manually picked using 20 μL pipette tips and dissociated to obtain neural stem cells/neural progenitor cells (NSCs/NPCs) using Accutase. NSCs/NPCs were then plated on Matrigel-coated 6-well plates and maintained in 3N media with 20 ng/mL bFGF2 and 20 ng/mL EGF. hiPSC-NSCs/NPCs were passaged every 4–5 days and cultured in the 37 °C incubator with a humidified atmosphere of 5% CO_2_ with daily medium change and then purified by passaging two more times with Accutase before cryopreservation or any usage for experiments. The induced hiPSC-NSCs/NPCs were confirmed by immunostaining with Nestin (NSC/NPCs marker) and Oct-3/4 (iPSCs marker) and showed over 99% positively stained hiPSCs-NSCs/NPCs by Nestin, and a lack of Oct-3/4 staining indicated that there were no remnant iPSCs in the culture (Appendix A).

### 4.5. Apoptosis Assay by Flow Cytometry in Human-Induced Pluripotent Stem Cells and Neural Stem Progenitor Cells (hiPSCs-NSCs/NPCs)

Siponimod/BAF312 and FTY720-p used in this study were supplied by Novartis Pharmaceuticals Corporation (Basel, Switzerland) and dissolved in DMSO solution. SWE2871 was purchased from Cayman Chemical (Ann Arbor, MI, USA). Human iPSCs-NSCs/NPCs were grown as adherent cultures in a 24-well plate in complete 3N medium with growth factors. Cultures were pretreated with indicated concentrations of siponimod/BAF312, FTY720-P, or the control DMSO with or without TNFα 20 ng/mL in minimum medium without growth factors for 48 h. On the assay day, the iPSC-NSCs/NPCs cells were then detached with Accutase digestion. After washing once with 3N medium, cells were resuspended in 1× Annexin binding buffer and stained with APC-conjugated Annexin V kit (BD Biosciences, Franklin Lakes, NJ, USA), 7-AAD viability staining solution (BioLegend, San Diego, CA, USA), and APC-conjugated Annexin V, and further analyzed using flow cytometry according to the manufacturer’s instructions. Surviving live cells are double-negative for both Annexin V and 7AAD. Relative cell death or survival was calculated compared to relevant vehicle controls. Staining was analyzed by flow cytometry and the apoptotic cell population was calculated by the percentage of Annexin V-positive cells using Flow Jo. The average of the duplicates was calculated as a fraction of the average of the vehicle control in that experiment, and then the average of four experiments of these fractions were presented as a column with SD. T-tests were performed to see significant differences. Statistical analysis was carried out by comparing each group to the vehicle control. * *p* < 0.05, ** *p* < 0.01 were considered statistically significant.

### 4.6. Live-Cell Apoptosis Analysis Using the IncuCyte System in Human-Induced Pluripotent Stem Cells and Neural Stem Progenitor Cells (hiPSCs-NSCs/NPCs)

To evaluate the effectiveness of siponimod/BAF312 on inducing cell apoptosis, hiPSCs-NSCs/NPCs were seeded in 96-well plates at 2.5 × 10^4^ cells/well overnight in complete medium with growth factors. The following day, when cells had reached a confluence of at least 75%, media without growth factors were replaced and hiPSCs-NSCs/NPCs were treated with 20 ng/mL of TNFα and varying concentrations of siponimod/BAF312 (0.01 nM, 0.1 nM, 0.5 nM, 1 nM, and 5 nM). To visualize the numbers of hiPSCs-NSCs/NPCs undergoing apoptosis, 5 µM of a caspase-3/7 reagent was also added to each well. Cells were then imaged at tenfold magnification in an IncuCyte^®^ S3 Live Cell Analysis System (Sartorius, Ann Arbor, MI, USA) at 37 °C with 5% CO_2_ for ~6 days. Images were acquired every 2 h, at five images per well. Data were analyzed using IncuCyte analysis software (v2020C) to detect and quantify the number of green (apoptotic) cells per image. A filter threshold of 75 µm^2^ was established to abolish green-fluorescent aberrations. The number of green events (apoptotic cells) was calculated by counting the caspase count per image, as previously published. Data were plotted using GraphPad Prism and Student’s *t*-test (2 tail) statistical analysis was used for further analysis. *p* < 0.05 was considered statistically significant.

### 4.7. RNA-Seq Analysis of Human-Induced Pluripotent Stem Cells and Neural Stem/Progenitor Cells (hiPSCs-NSCs/NPCs)

Bulk RNA-Seq was performed by Novogene. Briefly, total RNA from the control minimal medium without TNFα or TNFα alone, or TNFα plus siponimod-treated hiPSCs-NSCs/NPCs cell samples, were extracted using an RNAeasy MiniPrep Kit (Qiagen, Germantown, MD, USA). Only samples with high RNA integrity were subjected to library preparation. Samples were then sequenced on an Illumina Hiseq platform and 125 bp/150 bp paired-end reads were generated. The index of the reference genome was built using Bowtie v2.2.3 and paired-end clean reads were aligned to the reference genome using TopHat v2.0.12. Differential gene expression analysis was performed using DESeq2 [38]. *p* values were adjusted using the Benjamini–Hochberg method. Corrected *p*-values of 0.05 and log2 (fold-change) of 1 were set as the threshold for significantly differential expression. Functional analysis and gene ontology (GO) term enrichments were performed using iPathway Guide (Advaita bioinformatics) [39].

### 4.8. Statistical Analysis

Student’s *t*-test or a non-parametric Mann–Whitney U test was used for comparisons between different groups. Statistically significant *p*-values between groups that were calculated are shown above the horizontal lines in the figures. For reproducibility purposes, experiments were repeated. All statistical analyses were performed using GraphPad Prism 8 software (GraphPad Software, Inc., La Jolla, CA, USA). *p*-values < 0.05 were considered statistically significant and highly significant at ** *p* < 0.01 or *** *p* < 0.001. Kaplan–Meier and cumulative risk of death curves were generated with StatView software 5.0 and the statistical significance of differences between cohorts of neurons was determined with the log-rank test.

## Figures and Tables

**Figure 1 ijms-25-02454-f001:**
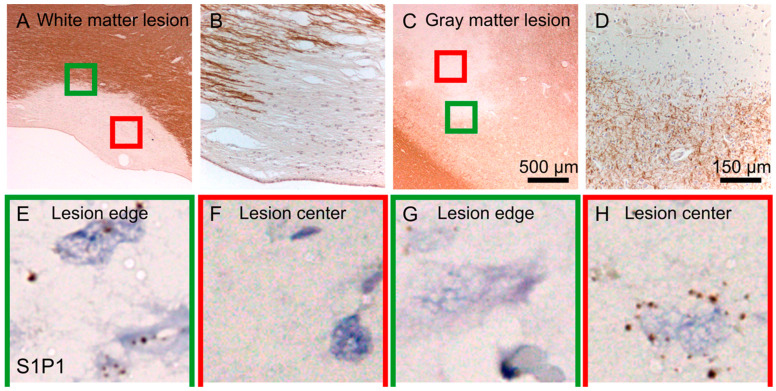
Sphingosine-1-phosphate (S1P) receptor 1 (S1P1R) is expressed in cortical demyelinating lesions of an MS brain. At a low magnification of a chronic active MS lesion (**A**), the green square is placed at the lesion border and the red square at the lesion center. At a higher magnification of the lesion border streaks of myelin (MBP, brown) are shown (**B**). At a low magnification of a cortical region containing an MS lesion (type III), the green square is placed at the lesion border and the red square within the lesion (**C**), with higher magnification of the lesion border (MBP, brown) (**D**). S1P1R is expressed in astrocytes at the border of active white matter lesions and less at the lesion center (**E**,**F**). In cortical lesions, S1P1R is expressed abundantly by neurons within the lesion (**G**,**H**).

**Figure 2 ijms-25-02454-f002:**
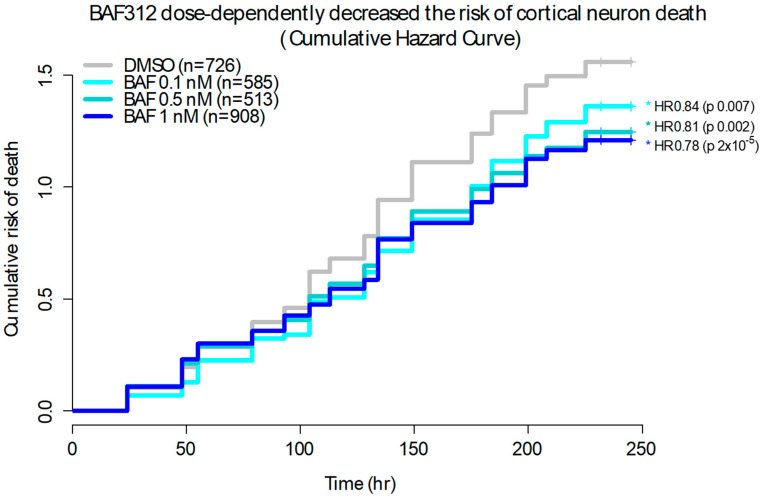
Siponimod/BAF312 dose-dependently enhanced neuronal survival in rat primary cultured cortical neurons. Rat cortical neurons were isolated and grown in a 96-well plate with neural basal medium. The neurons were transfected with EGFP, then treated with DMSO (n = 726) or Siponimod/BAF312 at 0.1 nM (n = 585), 0.5 nM (n = 513), or 1.0 nM and imaged. Thousands of individual rat cortical neurons were labelled and tracked. The cumulative hazard curves were plotted, displaying the risk of death for the cortical neurons from one representative experiment. Hazard Ratio (HR) and associated *p* value were calculated using COX proportional hazards analysis. HR < 1 denotes protection. In the plot, the *y*-axis represents a quantitative measure of the accumulated risk of cell death over time. Log-rank test was used to determine differences among different concentration of Siponimod/BAF312, with * *p* < 0.05 considered statistically significant.

**Figure 3 ijms-25-02454-f003:**
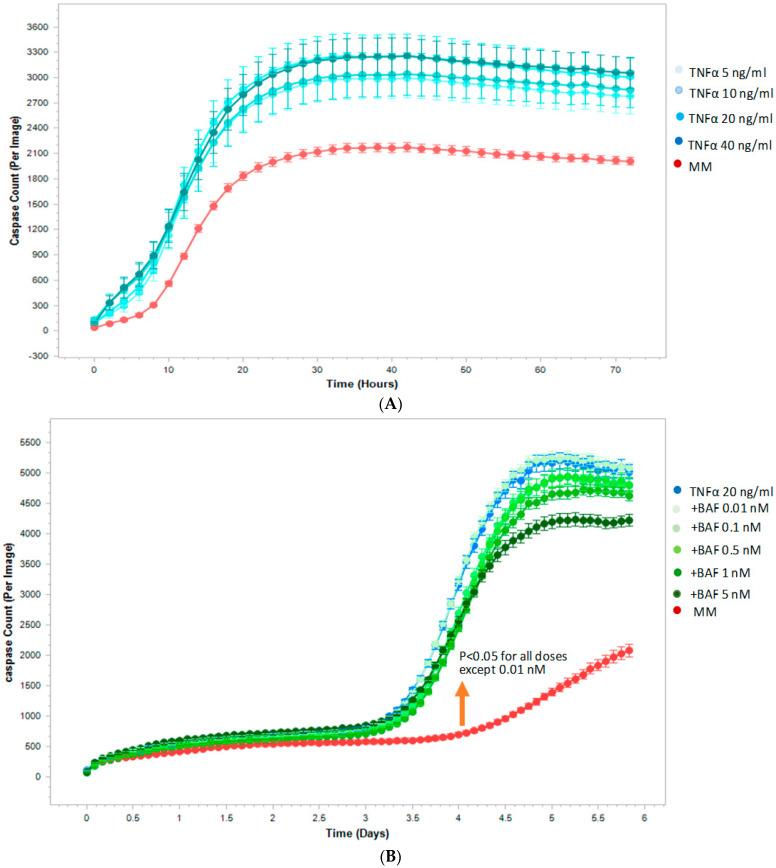
Siponimod/BAF312 effect on decreasing apoptosis in human-induced pluripotent stem cells and neural stem progenitor cells (hiPSCs-NSCs/NCPs) using the IncuCyte live-cell analysis system. hiPSCs-NSCs/NPCs cells were seeded in 96-well plates and treated with or without TNFα (20 ng/mL) under varying concentrations of siponimod/BAF312. Then, 5 µM of a caspase-3/7 reagent was added to each well to visualize the numbers of hiPSCs-NSCs/NPCs cells undergoing apoptosis. Cells were then imaged for ~6 days. Data were analyzed using IncuCyte analysis software (v2020C) to detect and quantify the number of green (apoptotic) cells per image. (**A**) TNFα dose response curve. TNFα dose-dependently (5, 10, 20, 40 ng/mL) increased apoptosis of hiPSCs-NSCs/NPCs cells (blue line) compared to medium alone without TNFα (red line). (**B**) Siponimod/BAF312 effect on decreasing apoptosis. Under the treatment of TNFα (20 ng/mL) (blue line), siponimod/BAF312 dose-dependently (0.1, 0.5, 1.0, 5.0 nM, except lowest dose of 0.01nM) decreased apoptosis of hiPSCs-NSCs/NPCs cells (green lines) compared to vehicle alone (TNFα 20 ng/mL, blue line) without siponimod/BAF312. (**C**) Enlarged scale to show statistical significance for all doses except the lowest dose of 0.01 nM that was sustained up to 5.8 days. These experiments were repeated three times with similar results. The data from one representative experiment are shown. Student’s *t*-test (2 tail) statistical analysis was used in these experiments. *p* values were set to ≤0.05 for statistical significance.

**Figure 4 ijms-25-02454-f004:**
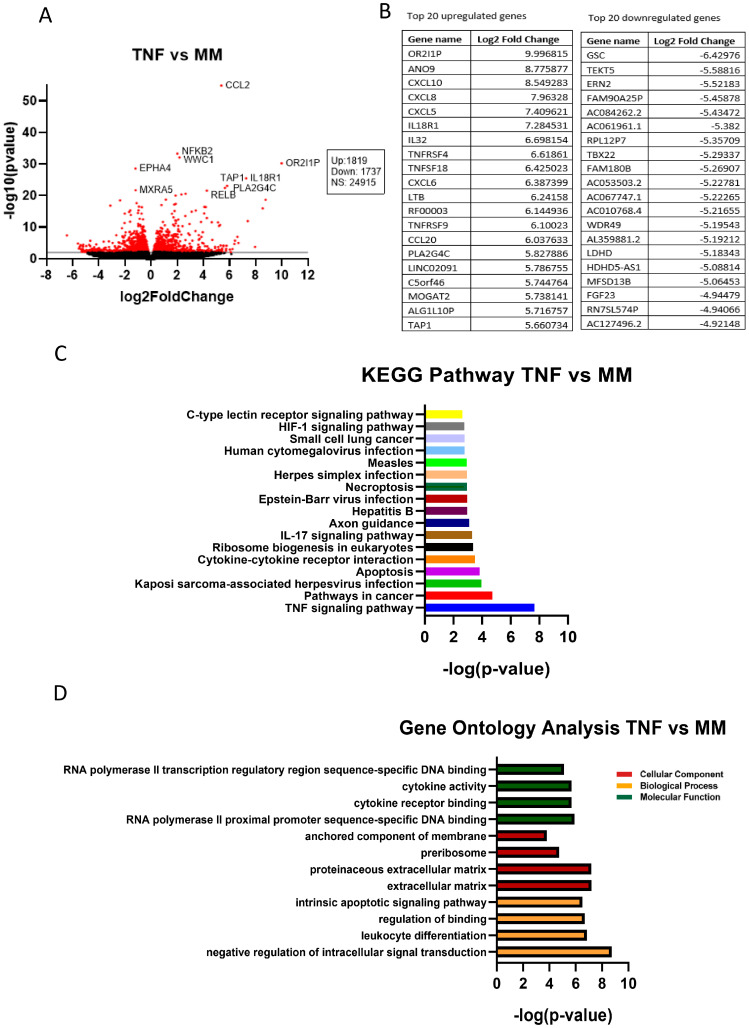
RNA-seq analysis of TNFα-treated human-induced pluripotent stem cells and neural stem progenitor cells (iPSCs-NSCs/NPCs) identifies important targets for cytokine–cytokine receptor interaction, apoptosis, and TNF signaling pathways. hiPSCs-NSCs/NPCs cells treated with or without TNFα (20 ng/mL) were used for RNA-seq analysis. (**A**) In the volcano plot, a total of 3556 genes were significantly differentially expressed by TNFα treatment compared to minimum media (MM) (red dots), with 1819 of them being upregulated and 1737 downregulated. (**B**) Tables show the top 20 upregulated (left) and downregulated (right) genes in TNFα-treated iPSCs-NSCs/NPCs. (**C**) The 18 most significantly enriched KEGG pathways after treatment with TNFα are shown. (**D**) Gene ontology analysis of the differentially expressed genes after TNFα treatment in hiPSCs-NSCs/NPCs cells. *n* = 5–6 sets.

**Figure 5 ijms-25-02454-f005:**
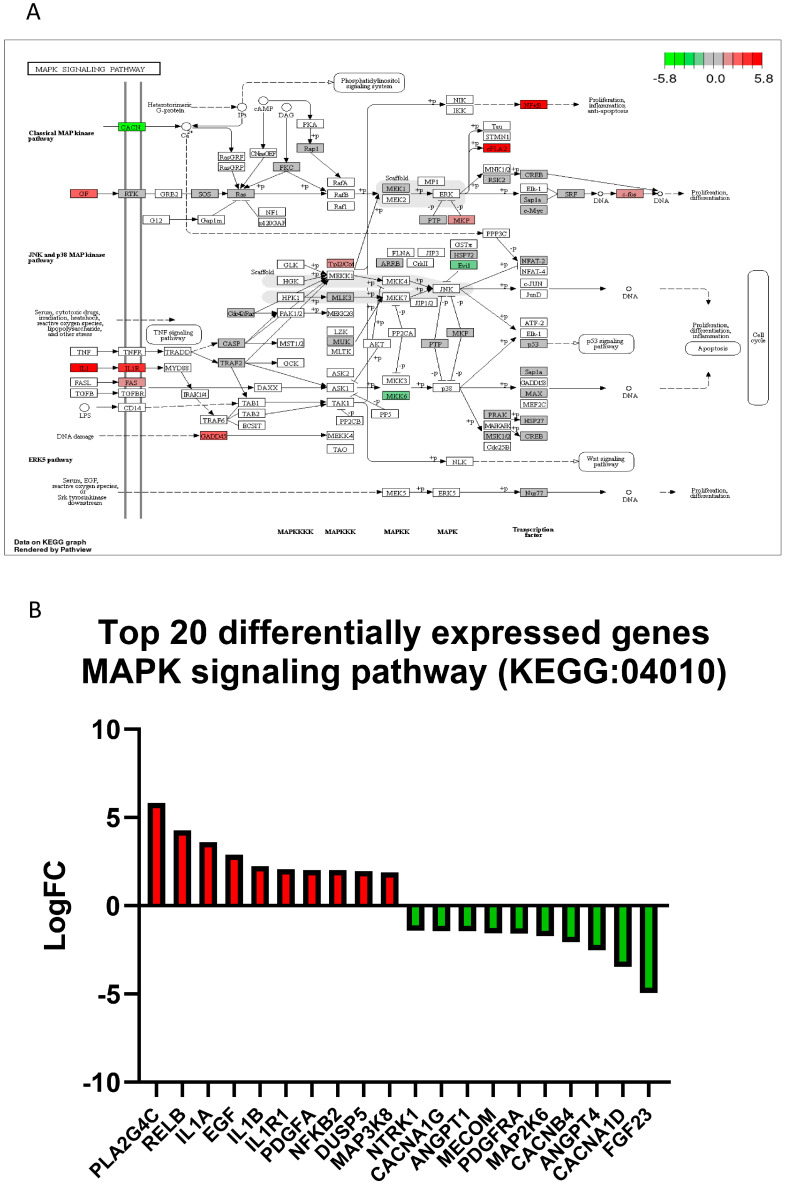
TNFα treatment affects expression profiles of MAPK signaling pathway genes in human-induced pluripotent stem cells and neural stem progenitor cells (hiPSCs-NSCs/NPCs). (**A**) The MAPK pathway (KEGG: 04010) diagram is overlaid with the expression changes of each gene. The legend describes the values on the gradient in LogFC. Downregulated genes are shown in green, while upregulated genes are in red. (**B**) The top 20 differentially expressed genes in the MAPK signaling pathway (KEGG: 04010) are ranked based on their absolute value of log fold change. Upregulated genes are shown in red, and downregulated genes are shown in green. *n* = 5–6 sets.

**Figure 6 ijms-25-02454-f006:**
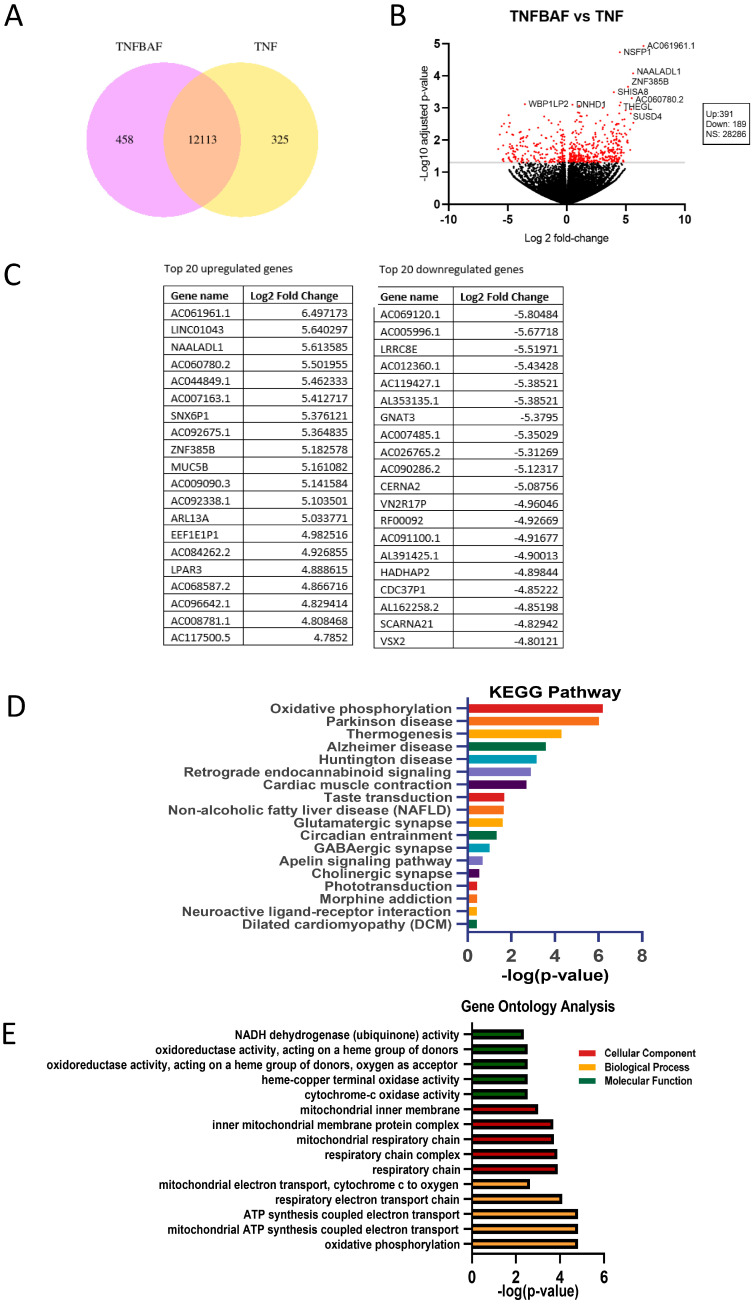
RNA-seq analysis of the effect of siponimod/BAF312 on TNFα-treated human-induced pluripotent stem cells and neural stem progenitor cells (hiPSCs-NSCs/NPCs) identifies important genes involved in oxidative phosphorylation. hiPSCs-NSCs/NPCs cells were used for mRNA-Seq analysis. (**A**) Pie chart of TNFBAF vs. TNF showing differential gene expression under TNFα + BAF312/siponimod (TNFBAF) treatment compared with TNFα alone. (**B**) In the volcano plot, a total of 580 genes were significantly differentially expressed by TNFBAF (0.1 nM) and TNFα (20 ng/mL) treatment in hiPSCs-NSCs/NPCs cells, with 391 upregulated genes and 189 downregulated genes. (**C**) Tables show the top 20 upregulated (left) and downregulated (right) genes. (**D**) The 18 most significantly enriched KEGG pathways after treatment with TNFBAF and TNFα are shown. (**E**) Gene ontology analysis of the differentially expressed genes after siponimod treatment in hiPSCs-NSCs/NPCs cells. *n* = 5–6 sets.

**Figure 7 ijms-25-02454-f007:**
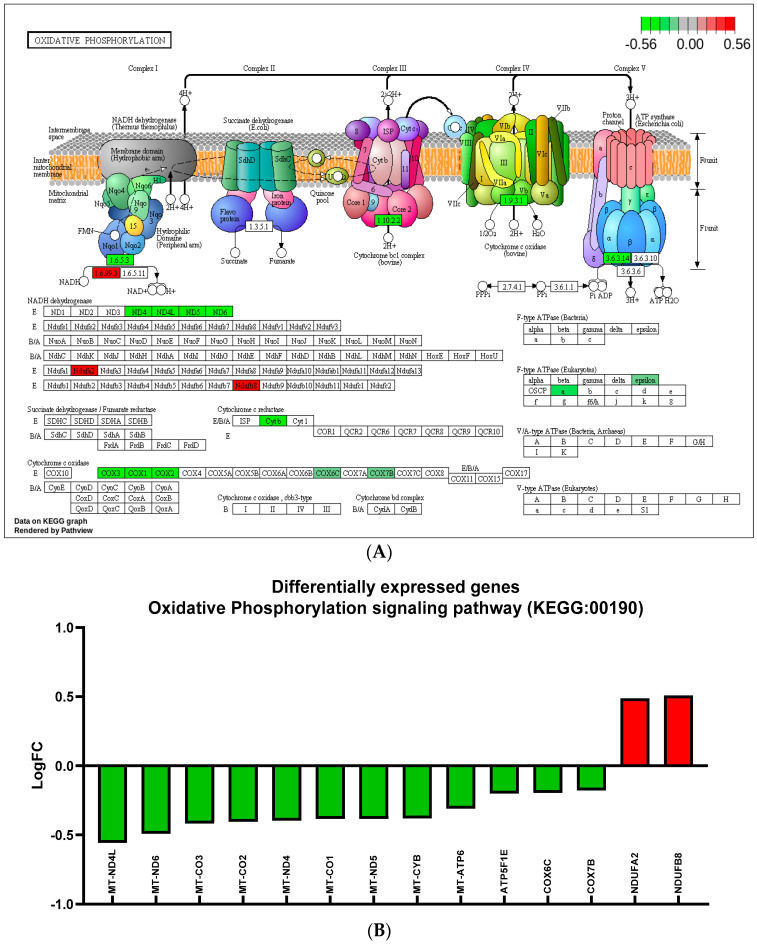
Siponimod/BAF312 treatment affects the oxidative phosphorylation signaling pathway in human neuronal hiPSCs-NSCs/NPCs cells. (**A**) The oxidative phosphorylation pathway (KEGG: 00190) diagram is overlaid with the expression changes of each gene. Siponimod/BAF312 treatment played a crucial role in regulating oxidative phosphorylation. LogFC was used for the degree of differential expression. Downregulated genes are shown in green, while upregulated genes are in red. (**B**) The top differentially expressed genes in oxidative phosphorylation pathway (KEGG: 00190) are ranked based on their absolute value of log fold change. Upregulated genes are shown in red, and downregulated genes are shown in green. *n* = 5–6 sets.

**Figure 8 ijms-25-02454-f008:**
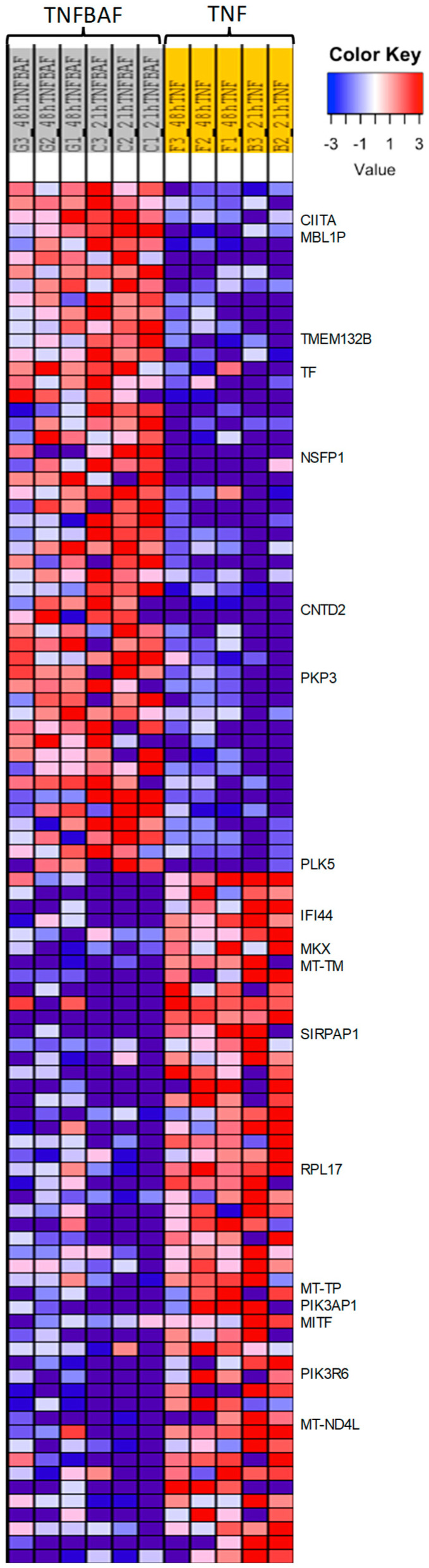
Heatmap of gene expression shows that siponimod/BAF312 treatment reverses TNFα-driven neuroinflammation in oxidative phosphorylation, apoptosis, MAPK/NFKB signaling, and cytokine to cytokine receptor interaction pathways in human neuronal hiPSCs-NSCs/NPCs cells. Heatmap shows differential expressed genes with siponimod/BAF312 treatment with TNFα as compared to TNFα alone in hiPSCs-NSCs/NPCs cells. Downregulated genes are shown in blue and upregulated genes are in red. Genes of interest are marked on the right, including genes involved in mitochondrial metabolic and inflammatory functions [MT-ND4L, a core subunit of the mitochondrial membrane respiratory chain NADH dehydrogenase (Complex I), which catalyzes electron transfer from NADH through the respiratory chain, using ubiquinone as an electron acceptor; MT-TM, the mitochondrially encoded tRNA methionine that transfers the amino acid methionine to a growing polypeptide chain at the ribosome site of protein synthesis during translation causing neuromuscular disorder mitochondrial myopathy; TMEM132B, a transmembrane protein that is overexpressed in the brain cortex and immune cells; and TF, where the transferrin is an iron-binding transport protein that is responsible for the transport of iron from sites of absorption and heme degradation to those of storage and utilization. Serum transferrin may have a role in stimulating cell proliferation as a growth factor that is involved in stem cell differentiation protocols towards the derivation of the following cells, like in neurons’ neural precursor-like cells, motor neuron progenitor cells, and motor neuron-like cells; PIK3AP1, the phosphoinositide-3-kinase adaptor protein 1 involved in the regulation of inflammatory response, regulation of signal transduction, and Toll-like receptor signaling pathway; MITF, a microphthalmia-associated transcription factor that regulates the expression of genes with essential roles in cell differentiation, proliferation, and survival; RPL17, the ribosomal protein L17 that is involved in peptide chain elongation and metabolism pathways; PIK3R6, a phosphoinositide-3-inase regulatory subunit 6 that is involved in metabolism pathways; MT-TP, Microsomal Triglyceride Transfer Protein, plays a central role in lipoprotein assembly; CIITA is class II major histocompatibility complex transactivator involved in the common mechanism of immune escape through reduction of MHC class II expression in primary mediastinal large B cell lymphoma; SIRPAP1 is signal regulatory protein α (SIRPα), a regulatory membrane glycoprotein from SIRP family expressed mainly by myeloid cells and also by stem cells or neurons; PLK5 is Polo Like Kinase 5 (Inactive) that is predicted to enable ATP binding activity and protein kinase activity involved in several processes, including defense response to tumor cell; positive regulation of neuron projection development; and regulation of G1/S transition of mitotic cell cycle; PKP3 (Plakophilin 3) expressed in dendritic reticular cells of lymphatic follicles; MKX (Mohawk Homeobox) is a transcription factor that promotes meniscus cell phenotype and tissue repair and reduces osteoarthritis severity. Smad3 binds it; IFI44 (Interferon Induced Protein 44) is an immune evasion biomarker for SARS-CoV-2 and Staphylococcus aureus infection and associated with Multisystem Inflammatory Syndrome; CNTD2 (cyclin N-terminal domain containing 2) is involved in sphingolipid metabolism; NSFP1 (N-ethylmaleimide-sensitive factor pseudogene 1) is widely expressed in brain tissue]. *n* = 5–6 pairs.

**Figure 9 ijms-25-02454-f009:**
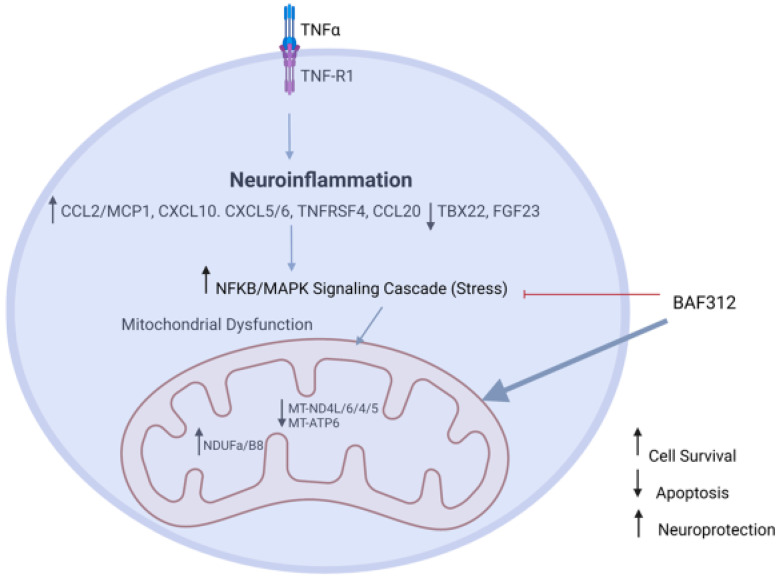
Diagram depicting the mechanism of neuroprotective effect of siponimod/BAF312. The potential mechanism of neuroprotective effect of siponimod/BAF312 is to reverse TNFα-driven neuroinflammation by affecting genes mainly involved in apoptosis, mitochondrial oxidative phosphorylation, MAPK/NFκB signaling pathways, and cytokine–cytokine receptor interaction. Siponimod/BAF312 decreases neuronal apoptosis through relieving oxidative stress in human neuronal cells, as shown in the schematic model.

## Data Availability

Data is contained within the article. Further details will be made available upon request.

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
