# Peer review of "Siponimod Attenuates Neuronal Cell Death Triggered by Neuroinflammation via NFκB and Mitochondrial Pathways"

_ijms, 2024, doi:10.3390/ijms25052454_

Round 1
Reviewer 1 Report (New Reviewer)
Comments and Suggestions for Authors
The reviewed manuscript addresses a very interesting and important topic of Siponimod influence on central nervous system cellular components. The influence of disease modifying treatment in MS on local CNS pathological processes belongs to the crucial scientific and clinical challenges of neurology. However, the reviewed manuscript in its current form contains multiple issues, which have to be clarified in order to enable the proper assessment.
The references have to be checked carefully – some examples: after number „5” references jump to number “9”, and this reference seems to be wrongly assigned at this place. Citation “10” refers to dimethyl fumarate and does not have anything in common with S1P modulators. Also the citation “22” seems to be wrongly assigned
Abbreviations should be properly introduced in the text and used consequently – there is an inconsistency regarding for example the use of iPSC / hiPSC / iPS / iPSCs-NPCs / hiPSCs-NSCs/NPCs
Numbers of experiments should be provided in each case
Figure legends should be also improved, better described, the meaning of bars and abbreviations should be explained.
Regarding neuronal survival in rat primary cultured cortical neurons, the authors should unify the information about the number of neurons fallowed in the experiment (hundreds – in the text or thousands – in the figure legend), the “n” should be explained
Why the concentrations of Siponimod used varied among different experiments
Why 0.1 nM concentration of Siponimod was used for transcriptomic analysis, although this was the concentration without anti-apoptotic effects in live cell analysis. What was the concentration of TNFa in theses experiments 10 or 20ng/ml? What does the abbreviation TNFBAF mean?
In the discussion: the authors mention that “In our current study, we also confirmed that the S1P1 receptor is expressed in both rat and human neuronal progenitor cells/stem cells and siponimod increased S1P1 receptor degradation in rat NPC neurospheres (NS)/rNPC and a human iPSC-NSCs/NPCs cell line” but such results are not reported at all.
The discussion is too scarce.
Comments on the Quality of English Language
none
Author Response
Dear Editor:
We would like to thank you and reviewers for your careful reading of the manuscript and your constructive suggestions. Please find below in bold a point-by-point response to reviewer #1:
- The references have to be checked carefully – some examples: after number “5” references jump to number “9”, and this reference seems to be wrongly assigned at this place. Citation “10” refers to dimethyl fumarate and does not have anything in common with S1P modulators. Also, the citation “22” seems to be wrongly assigned.
We have reviewed all the references and made sure that 1) references are listed chronologically, and 2) the scientific papers are cited correctly. In addition, we have eliminated former citations 10 and 22, as they were incorrectly cited in the first version of the manuscript.
- Abbreviations should be properly introduced in the text and used consequently – there is an inconsistency regarding for example the use of iPSC / hiPSC / iPS / iPSCs-NPCs / hiPSCs-NSCs/NPCs.
We have now properly introduced those abbreviations in method section (P6-8) and in the text for each and every section. We hope that these changes, highlighted in yellow, make our manuscript easier to read. We also changed the title of several figures (3, 4, 5 and 6) to avoid further issues. (E.g., from “Human iPSCs-NSCs/NPCs transcriptomic analysis of TNFα effects” to “Transcriptomic analysis of TNFα treated human iPSCs-NSCs/NPCs)”.
iPSC: induced pluripotent stem cells.
hiPSC: human induced pluripotent cells.
iPSCs-NPCs: induced pluripotent stem cells – derived neural progenitor cells.
hiPSCs-NSCs/NPCs: human induced pluripotent stem cell-derived neural stem/progenitor cells.
- Numbers of experiments should be provided in each case.
We thank the reviewer for pointing out this issue. We have added additional information regarding the number of times that each experiment was repeated in this study. These changes can be seen both in the methodology section and in the figure legends.
- Figure legends should be also improved, better described, the meaning of bars and abbreviations should be explained.
We have revised the figure legends to include more details regarding abbreviations used, units, and statistical analysis.
- Regarding neuronal survival in rat primary cultured cortical neurons, the authors should unify the information about the number of neurons fallowed in the experiment (hundreds – in the text or thousands – in the figure legend), the “n” should be explained.
In figure 2, we can confirm that the number of rat primary cultured cortical neurons labelled and tracked was in the thousands range. We have made sure that such changes are reflected in the manuscript.
- Why the concentrations of Siponimod used varied among different experiments. Why 0.1 nM concentration of Siponimod was used for transcriptomic analysis, although this was the concentration without anti-apoptotic effects in live cell analysis. What was the concentration of TNFa in these experiments 10 or 20ng/ml? What does the abbreviation TNFBAF mean?
As stated throughout the paper, the concentration of siponimod/BAF312 used in the transcriptomic experiments was 0.1 nM. Importantly, in the IncuCyte live cell analysis, we found that with the exception of the lowest dose (0.01 nM), siponimod/BAF312 significantly decreased the number of apoptotic cells (Figure 3B, 3C). Therefore, we decided to use 0.1 nM for the subsequent transcriptomic experiments.
The concentration of TNFa used in the transcriptomic experiments was 20ng/ml. We have made it more clear in the manuscript.
The abbreviation TNFBAF refers to the combination of TNFa and BAF312 (Siponimoid) and it means that the cells were treated with both TNFa and BAF312. We have clarified this potential issue in the manuscript.
- In the discussion: the authors mention that “In our current study, we also confirmed that the S1P1 receptor is expressed in both rat and human neuronal progenitor cells/stem cells and siponimod increased S1P1 receptor degradation in rat NPC neurospheres (NS)/rNPC and a human iPSC-NSCs/NPCs cell line” but such results are not reported at all.
We have deleted that paragraph, as suggested by the reviewer.
- The discussion is too scarce.
We have added more relevant information to the conclusion.
We again thank you and the reviewers for your insightful comments. We hope we have answered the reviewers’ concerns satisfactorily in the current version.
Sincerely,
Yang Mao-Draayer, MD, PhD
Professor in Neurology
Director of Clinical and Experimental Therapeutics
Multiple Sclerosis Center of Excellence
Arthritis and Clinical Immunology Research Program
Oklahoma Medical Research Foundation
820 NE 15th Street
Oklahoma City, OK 73104
Email: Yang-Mao-Draayer@omrf.org
Reviewer 2 Report (New Reviewer)
Comments and Suggestions for Authors
Siponimod Attenuates Neuronal Cell Death Triggered by Neuroinflammation via NFkB and Mitochondrial Pathways.
The primary objective of this study was to establish the neuroprotective properties of Siponimod in both rat cortical neurons and human neuronal cells by modulating S1P receptors. The investigation focused on its potential implications for multiple sclerosis, specifically in addressing the demyelination of neurons. Neuronal inflammation was induced using TNF-α as a model in order to assess the efficacy of Siponimod. Rigorous and highly effective methodologies were employed to successfully achieve the predetermined objectives. Additionally, the study validated the significance of S1P1 expression in demyelinating lesions through the analysis of central nervous system autopsy tissues obtained from individuals with the targeted condition. The Kaplan-Meier survival analysis provided evidence for a decreased risk of cortical neuronal death following Siponimod treatment in rats. Additionally, the results of the pro-survival analysis elucidated a reduction in TNF-α induced apoptosis in human neuronal iPSCs-NSCs/NPCs cells attributed to Siponimod. Moreover, the research was expanded through RNA-Seq analysis to investigate the impact of Siponimod on TNF-α-driven inflammatory gene expression. This extension aimed to identify specific targets crucial for oxidative phosphorylation in human neuronal hiPSCs-NSCs/NPCs cells. Finally, the application of a gene expression heatmap adeptly demonstrated that Siponimod treatment effectively counteracted TNF-α-driven neuroinflammation in human neuronal hiPSCs-NSCs/NPCs cells. This reversal was evident across key pathways, encompassing oxidative phosphorylation, apoptosis, MAPK/NFKB signaling, and cytokine-to-cytokine receptor interaction.
The study was conducted with excellence, and the results were thoroughly elucidated. Based on my comprehensive exploration, I find no further recommendations for improvements.
Author Response
Dear Editor:
We would like to thank reviewer #2 for their careful reading of the manuscript.
Reviewer #2 did not have any concerns.
Sincerely,
Yang Mao-Draayer, MD, PhD
Professor in Neurology
Director of Clinical and Experimental Therapeutics
Multiple Sclerosis Center of Excellence
Arthritis and Clinical Immunology Research Program
Oklahoma Medical Research Foundation
820 NE 15th Street
Oklahoma City, OK 73104
Email: Yang-Mao-Draayer@omrf.org
This manuscript is a resubmission of an earlier submission. The following is a list of the peer review reports and author responses from that submission.
Round 1
Reviewer 1 Report
Comments and Suggestions for Authors
I think I have a chance to read the interesting article before publishing, well done.
There are some points:
1- It should better analysis the data which repeated over time with Repeated measure of ANOVA.
2- Please report the result of analysis based on APA style.
Reviewer 2 Report
Comments and Suggestions for Authors
In the article entitled “Siponimod Attenuates Neuronal Cell Death Triggered by Neuroinflammation via NFkB and Mitochondrial Pathways”, Qin Wang and colleagues wants to demonstrate the effect of Siponimod on neurons cell death.
The way the paper is written is not suitable for the presentation of a scientific article, and it lacks clarity and a precise flow.
The introduction is too concise and does not provide sufficient background.
Material and methods section is not properly reported. Also, it is not clear why the authors reported information about statistical significance at the end of each method section.
It is not clearly stated how the results regarding the GFP labeled neurons were extrapolated and used to make the survival analysis.
Several parts are also mismatching compared to the main hypothesis of the study. For example, the authors state that they want “to evaluate the effectiveness of siponimod on inducing cell apoptosis” at line 174-175, while they previously state that Siponimod have neuroprotective effects.
It is not clear why the authors consider a western blot assay as S1P1 receptor degradation assay.
It is not clear based on what assumption the authors choose to use parametric or non-parametric test.
The results are not properly reported as well. Also, the figure quality is very low, and it is difficult to read them since their blurriness (especially figure 1, 3, 4, 8, suppl. Figure 3).
In figure 1, the green and red squares insets do not seem to correspond to their image of origin. Also, the low quality of the picture does not allow a clear comprehension of the results.
Regarding “mRNA expression of the S1P1 receptor in demyelinating lesions of CNS autopsy tissue from an MS patient” paragraph: how can the authors know that they are astrocytes if no specific markers (e.g., GFAP) are showed?
Regarding “Siponimod/BAF312 dose-dependently enhanced neuronal survival in rat primary cultured cortical neurons” paragraph: it is not clearly reported how the authors followed hundreds of neurons, how they considered them live or dead cells and how the obtained results were used to create the figure 2.
Regarding “Siponimod/BAF312 increased S1P1 receptor degradation” paragraph: why the authors say that siponimod/BAF312 degraded S1P1 receptor in rat NPC neurosphere if in the picture no significance is reported. Also, no standard deviation is showed, and that means that only one sample was used for the analysis; that is not sufficient for a scientific article. Also, the authors say that a dose-dependent effect was seen for the results in fig. 3, panel B. However, no significance is reported, and also the histograms do not show an actual dose-dependent response.
For fig 3, panel C, no histogram are included.
It is not clear what kind of statistical analysis was used to evaluate the data reported in figure 4. How the authors considered the two independent variables (i.e., time and treatments) to interpret the results?
RNA-seq data are not clearly reported and such data are not well integrated and discussed in the paper with the other results.
The discussion does not include and integrates the amount of different data contained in the study. The conclusions are not really supported by the results.
In summary, we are sorry to conclude that the present article is not suitable for publication.